# Bayesian network predicted variables for good neurological outcomes in patients with out-of-hospital cardiac arrest

**Kota Shinada**◉*, **Ayaka Matsuoka, Hiroyuki Koami**◉, **Yuichiro Sakamoto**

Department of Emergency and Critical Care Medicine, Faculty of Medicine, Saga University, Saga City, Saga Prefecture, Japan

* st9137@cc.saga-u.ac.jp

## Abstract

Out-of-hospital cardiac arrest (OHCA) is linked to a poor prognosis and remains a public health concern. Several studies have predicted good neurological outcomes of OHCA. In this study, we used the Bayesian network to identify variables closely associated with good neurological survival outcomes in patients with OHCA. This was a retrospective observational study using the Japan Association for Acute Medicine OHCA registry. Fifteen explanatory variables were used, and the outcome was one-month survival with Glasgow–Pittsburgh cerebral performance category (CPC) 1–2. The 2014–2018 dataset was used as training data. The variables selected were identified and a sensitivity analysis was performed. The 2019 dataset was used for the validation analysis. Four variables were identified, including the motor response component of the Glasgow Coma Scale (GCS M), initial rhythm, age, and absence of epinephrine. Estimated probabilities were increased in the following order: GCS M score: 2–6; epinephrine: non-administered; initial rhythm: spontaneous rhythm and shockable; and age: <58 and 59–70 years. The validation showed a sensitivity of 75.4% and a specificity of 95.4%. We identified GCS M score of 2–6, initial rhythm (spontaneous rhythm and shockable), younger age, and absence of epinephrine as variables associated with one-month survival with CPC 1–2. These variables may help clinicians in the decision-making process while treating patients with OHCA.

## Introduction

Out-of-hospital cardiac arrest (OHCA) is a public health concern and a condition with poor prognosis [1, 2]. Accurate prognostic prediction of OHCA is important for appropriate resource allocation for emergency medicine and for providing appropriate information to families [1, 3]. Various prediction models have been attempted for a variety of situations [4], including survival prediction [5–9] and good neurological prognosis [10, 11] for patients in whom return of spontaneous circulation (ROSC) has been achieved or target temperature management therapy has been initiated. In recent years, machine learning models have been developed and validated [12–15], further improving the accuracy of OHCA prognosis prediction.

**Data Availability Statement:** The data are owned by a third party. Data are available from the JAAM-OHCA registry committee (contact via http://www.

jaamohca-web.com/) for researchers who meet the criteria for access to confidential data.

**Funding:** The authors received no specific funding for this work.

**Competing interests:** The authors have declared that no competing interests exist.

Concomitantly, the risks of making clinical decisions based solely on prognostic models to determine the course of treatment for patients with OHCA have been discussed. Clinical decisions made according to prognostic models are not always accurate, and there is a risk of withholding treatment in potentially life-saving situations if incorrect decisions are made [16]. Furthermore, to use a predictive model, all components included in the model must be in place at the time the predictive model is used. In other words, if even one of the components of the predictive model is not present, the model may not be usable. Variables that are associated with a favorable prognosis have been reported [17]. However, which variables more directly predict a good prognosis has not been clarified. Bayesian networks build graphical models of causal relationships between events based on uncertain information and calculate the probability that the event they wish to estimate will occur from the given information [18, 19]. Compared with other deep learning methods, Bayesian network allows visualization of the relationships among factors and offers high explanatory potential [20]. This method has been widely applied in medicine, primarily in the fields of cardiac, cancer, psychiatric, and pulmonary diseases [21]. In our facility, we employ BayoLinkS (NTT DATA Mathematical Systems Inc., Tokyo, Japan) to estimate the prognosis of emergency patients and for clinical applications.

In this study, we used a Bayesian network to search for variables associated with the event of good neurological prognosis in adult patients with OHCA who had achieved ROSC.

## Materials and methods

### Study design and participants

This was a retrospective observational study using the Japan Association for Acute Medicine (JAAM) OHCA registry, a prospective observational data registry kept by JAAM, with participating facilities across Japan. The registry was launched on June 1, 2014 and is still accumulating data. As of January 2023, 99 hospitals from 37 of the 47 prefectures in Japan are included in the registry. JAAM OHCA registry collects data following a patient's arrival at the hospital (available Japanese item from: http://www.jaamohca-web.com/download/. Accessed 1st August 2022). The data individually entered by the hospital is checked by the JAAM OHCA registry committee of experts in emergency medicine and clinical epidemiology, who also perform data cleansing. Moreover, the data were combined with the pre-hospital data from the All-Japan Utstein Registry of the Fire and Disaster Management Agency [22–24].

Patients not resuscitated in the hospital, not linked to pre-hospital records, exogenous cardiac arrest cases, patients who had not achieved ROSC, and patients aged <18 years were excluded. Moreover, cases with missing appropriate data regarding no flow and low flow time (positive value and <400 minutes, respectively), epinephrine administration, GCS M score, blood gas test, and biochemistry test results were also excluded. One-month survival data is routinely collected in both the Fire and Disaster Management Agency Utstein Registry and Japan Association for Acute Medicine OHCA Registry, and there were no cases with missing information.

This study was approved by the Ethics Committee of the Saga University Hospital (Approval no. 2021-04-R-08) and conforms to the tenets of the Declaration of Helsinki. The need for informed consent was waived owing to the retrospective nature of the study.

### Variables and outcome

Fifteen variables were used based on previous studies [11, 25–32]: cause of cardiac arrest, age, sex, presence of bystander CPR, presence of witnesses/no flow time, initial emergency medical services (EMS) rhythm, presence of epinephrine administration, low flow time, motor

response in the Glasgow coma scale (GCS M), blood gas test results (pH, lactate, glucose) taken after ROSC from the emergency room to admission to the intensive care unit (ICU), and biochemical test results (creatinine, albumin, potassium) taken after the first hospital arrival. The outcome was one-month survival with Glasgow–Pittsburgh cerebral performance category (CPC) 1–2.

### Identification of variables closely associated with one-month survival with CPC1-2

The 2014–2018 and 2019 datasets were used as training and validation data, respectively. The following data were used in the analysis in a non-regressive order: cause of cardiac arrest, age and sex, presence of bystander CPR, presence of witnesses/no flow time (time from witnessing to start of CPR), EMS initial rhythm, presence of epinephrine administration, low flow time (time from start of CPR to ROSC), GCS M, and blood gas test and biochemical test results. Variables involved in one-month survival with CPC 1–2 were selected based on the training data, which were subsequently used in the sensitivity analysis.

### Statistical analysis

Patient characteristics were analyzed using JMP Pro version 14 (SAS Inc., Cary, NC, USA). Blood test results were divided into three groups using reference values: below reference values, within reference values, and below reference values. The reference values for glucose, creatinine, albumin, and potassium were taken from https://www.jslm.org/books/guideline/2021/GL2021_04.pdf, whereas those for pH and lactate were taken from https://www.acute-care.jp/ja-jp/learning/glossary/bloodgas (both accessed on July 1st, 2023). Except for blood tests, continuous variables were transformed into categorical variables using quartiles. All variables are presented as counts, followed by percentages in parentheses. The comparisons between the training data and the test one were made using the chi-square test. P<0.05 was considered significant. BayoLinkS was used to build and validate the Bayesian network model as well as for the sensitivity analysis.

### Results

Of the 57,754 cases enrolled in the study period, 5,340 were included in the analysis; of these, 4,286 and 1,054 cases were used as training and validation sets, respectively (Fig 1). The baseline characteristics and cardiac arrest details are described in Table 1. The training data showed significantly higher levels of low flow time (>39 minutes), lactate (>12.1 mg/dL), creatinine (<0.48 and 0.49–1.08 mg/dL), and albumin (<4.0 g/dL) and significantly lower levels of lactate (5.0–12.0 mg/dL), creatinine (>1.09 mg/dL), and albumin (4.1–5.1 g/dL) than those from the validation data. No significant differences were found for the other items.

Four variables, including GCS M, initial rhythm, age, and absence of epinephrine were chosen for one-month survival with CPC 1–2 in the training model (Fig 2). The estimated probabilities for each combination are presented in S1 Table.

The results of the sensitivity analysis are shown in Table 2. The estimated probabilities increased in the following order: GCS M score: 2–6; epinephrine: non-administered; initial rhythm: spontaneous rhythm and shockable; and age: <58 and 59–70. In contrast, they decreased in the following order: initial rhythm: asystole; age: 71–80 and >81; epinephrine: administered; initial rhythm: pulseless electrical activity; and GCS M score: 1. The validation analysis showed a sensitivity of 75.4% and a specificity of 95.4% (Table 3).

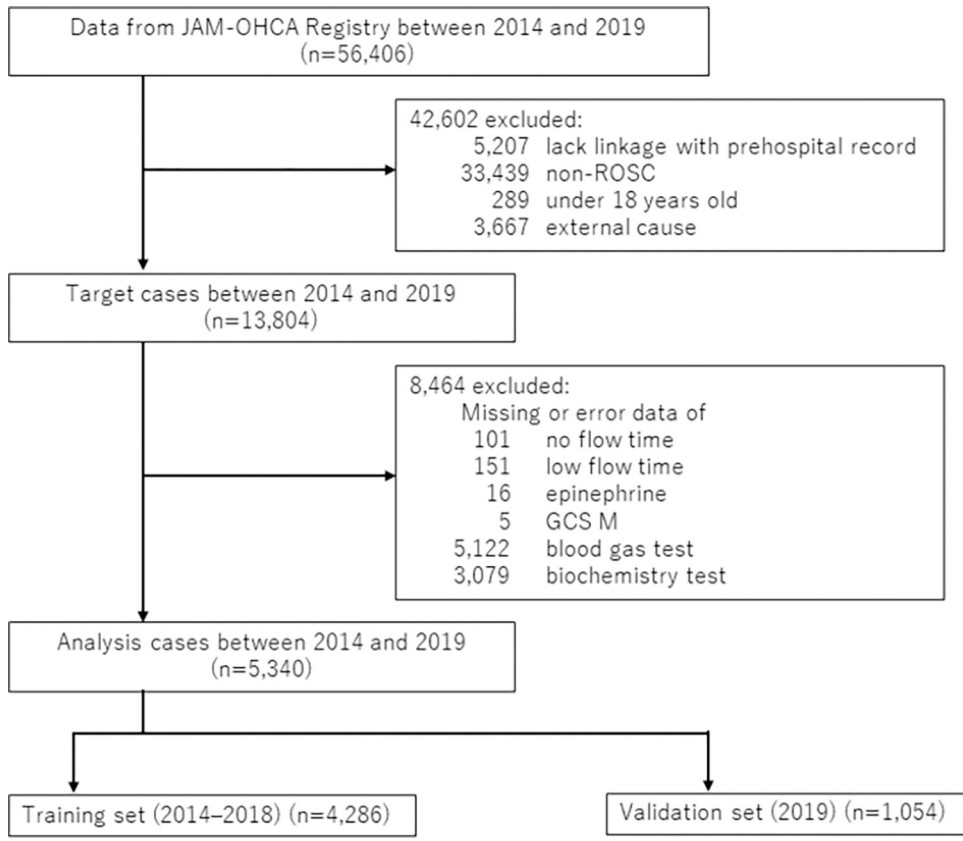

**Fig 1. Flow diagram of the patient selection procedure.** GSC M, motor response in the Glasgow coma scale; JAAM, Japan association for acute medicine; OHCA, out-of-hospital cardiac arrest; ROSC, return of spontaneous circulation.

## Discussion

We used a Bayesian network to identify variables associated with good neurological prognosis in adult patients with OHCA who had achieved ROSC and visualize the relationships among the variables. The variables included GCS M score after ROSC, initial rhythm, age, and absence of epinephrine, all of which have been used as components of previous OHCA prognostic variable exploration studies and predictive models (S2 Table).

Some of the predictive models that have been developed and studied to date are highly accurate and have been tested for practicality [4]. For example, the NULL-PLEASE score reported in 2017 [9] has been frequently validated as a prognostic model for OHCA, suggesting that it may perform better than other models [4, 33]. Modifications of the NULL-PLEASE have also been attempted to create models with fewer components [34]. However, the results do not always indicate a good prognosis. Previously, Kjetil et al. argued that a high degree of accuracy is required when considering predictive models for OHCA; however, clinical decisions based solely on predictive models also carry the risk of overlooking potentially life-saving situations [16]. Therefore, we believe that it is important to encourage clinicians to make comprehensive judgments by specifying the priority of variables. This study identified four variables that can be adapted to patients after ROSC and lead to a good neurological prognosis. Factors leading to the four variables were also identified from the Bayesian network model. Knowledge of these favorable prognostic variables may help clinicians to decide which tests and treatments to offer to patients and effectively communicate with their families.

**Table 1. Characteristics of the study population.**

| Variable | | All (n = 5,340) | Training data (n = 4,286) | Test data (n = 1,054) | P value |
|---|---|---|---|---|---|
| **Cause** | | | | | |
| | **Cardiac** | 3,553 (66.5%) | 2,835 (66.2%) | 718 (68.1%) | 0.2294 |
| | **Noncardiac** | 1,787 (33.5%) | 1,451 (33.9%) | 336 (31.9%) | 0.2294 |
| **Age** | | | | | |
| | **<58 years** | 1,297 (24.3%) | 1,039 (24.2%) | 258 (24.5%) | 0.8727 |
| | **59–70 years** | 1,315 (24.6%) | 1,078 (25.2%) | 237 (22.5%) | 0.0727 |
| | **71–80 years** | 1,354 (25.4%) | 1,078 (25.2%) | 276 (26.2%) | 0.5018 |
| | **>81 years** | 1,374 (25.7%) | 1,091 (25.5%) | 283 (26.9%) | 0.3657 |
| **Sex (Female)** | | 1,770 (33.1%) | 1,427 (33.3%) | 343 (32.5%) | 0.6613 |
| **Bystander CPR** | | 2,580 (48.3%) | 2,068 (48.3%) | 512 (48.6%) | 0.8635 |
| **Bystander defibrillation** | | 376 (7.0%) | 308 (7.2%) | 68 (6.5%) | 0.4209 |
| **No flow time / Unwitnessed** | | | | | |
| | **0 minutes** | 1,340 (25.1%) | 1,071 (25.0%) | 269 (25.5%) | 0.7213 |
| | **1–2 minutes** | 630 (11.8%) | 496 (11.6%) | 134 (12.7%) | 0.3113 |
| | **3–7 minutes** | 877 (16.4%) | 703 (16.4%) | 174 (16.5%) | 0.9261 |
| | **>8 minutes** | 937 (17.5%) | 767 (17.9%) | 170 (16.1%) | 0.1899 |
| | **Unwitnessed** | 1,556 (29.1%) | 1,249 (29.1%) | 307 (29.1%) | 1.0000 |
| **Initial rhythm** | | | | | |
| | **Shockable** | 1,548 (29.0%) | 1,260 (29.4%) | 288 (27.3%) | 0.1976 |
| | **Pulseless electrical activity** | 1,632 (30.6%) | 1,304 (30.4%) | 328 (31.1%) | 0.6815 |
| | **Asystole** | 1,577 (29.5%) | 1,261 (29.4%) | 316 (30.0%) | 0.7345 |
| | **Spontaneous rhythm** | 583 (10.9%) | 461 (10.8%) | 122 (11.6%) | 0.4406 |
| **Epinephrine** | | 4,047 (75.8%) | 3,251 (75.9%) | 796 (75.5%) | 0.8410 |
| **Physician-staffed EMS** | | 973 (18.2%) | 808 (18.9%) | 165 (15.7%) | 0.0161 |
| **Extracorporeal CPR** | | 850 (15.9%) | 680 (15.9%) | 170 (16.1%) | 0.8509 |
| **IABP** | | 875 (16.4%) | 714 (16.7%) | 161 (15.3%) | 0.2857 |
| **CAG** | | 1,896 (35.5%) | 1,519 (35.4%) | 377 (35.8%) | 0.8575 |
| **PCI** | | 926 (17.3%) | 752 (17.6%) | 174 (16.5%) | 0.4404 |
| **TTM** | | 1,545 (28.9%) | 1,251 (29.2%) | 294 (27.9%) | 0.4260 |
| **Low flow time** | | | | | |
| | **<13 minutes** | 1,255 (23.5%) | 984 (23.0%) | 271 (25.7%) | 0.0622 |
| | **14–24 minutes** | 1,312 (24.6%) | 1,040 (24.3%) | 272 (25.8%) | 0.2993 |
| | **25–38 minutes** | 1,431 (26.8%) | 1,156 (27.0%) | 275 (26.1%) | 0.5869 |
| | **>39 minutes** | 1,342 (25.1%) | 1,106 (25.8%) | 236 (22.4%) | 0.0238 |
| **GCS M score** | | | | | |
| | **1** | 4,720 (88.4%) | 3,791 (88.5%) | 929 (88.1%) | 0.7885 |
| | **2–6** | 620 (11.6%) | 495 (11.6%) | 125 (11.9%) | 0.7885 |
| **pH** | | | | | |
| | **<7.349** | 4,812 (90.1%) | 3,874 (90.4%) | 938 (89.0%) | 0.1851 |
| | **7.350–7.450** | 432 (8.1%) | 342 (8.0%) | 90 (8.5%) | 0.5704 |
| | **>7.451** | 96 (1.8%) | 70 (1.6%) | 26 (2.5%) | 0.0708 |
| **Lactate** | | | | | |
| | **<4.9 mg/dL** | 37 (0.7%) | 25 (0.6%) | 12 (1.1%) | 0.0613 |
| | **5.0–12.0 mg/dL** | 117 (2.2%) | 73 (1.7%) | 44 (4.2%) | <0.0001 |
| | **>12.1 mg/dL** | 5,186 (97.1%) | 4,188 (97.7%) | 998 (94.7%) | <0.0001 |
| **Glucose** | | | | | |
| | **<72 mg/dL** | 271 (5.1%) | 224 (5.2%) | 47 (4.5%) | 0.3472 |

*(Continued)*

**Table 1.** (Continued)

| Variable | | All (n = 5,340) | Training data (n = 4,286) | Test data (n = 1,054) | P value |
|---|---|---|---|---|---|
| | 73–109 mg/dL | 259 (4.9%) | 207 (4.8%) | 52 (4.9%) | 0.8730 |
| | >110 mg/dL | 4,810 (90.1%) | 3,855 (89.9%) | 955 (90.6%) | 0.5653 |
| Creatinine | | | | | |
| | <0.48 mg/dL | 57 (1.1%) | 53 (1.2%) | 4 (0.4%) | 0.0116 |
| | 0.49–1.08 mg/dL | 2,153 (40.3%) | 1,764 (41.2%) | 389 (36.9%) | 0.0117 |
| | >1.09 mg/dL | 3,130 (58.6%) | 2,469 (57.6%) | 661 (62.7%) | 0.0027 |
| Albumin | | | | | |
| | <4.0 g/dL | 4,797 (89.8%) | 3,869 (90.3%) | 928 (88.1%) | 0.0353 |
| | 4.1–5.1 g/dL | 540 (10.1%) | 414 (9.7%) | 126 (12.0%) | 0.0301 |
| | >5.2 g/dL | 3 (0.1%) | 3 (0.1%) | 0 (0.0%) | 1.0000 |
| Potassium | | | | | |
| | <3.5 mmol/L | 1,077 (20.2%) | 860 (20.1%) | 217 (20.6%) | 0.7000 |
| | 3.6–4.8 mmol/L | 2,205 (41.3%) | 1,795 (41.9%) | 410 (38.9%) | 0.0809 |
| | >4.9 mmol/L | 2,058 (38.5%) | 1,631 (38.1%) | 427 (40.5%) | 0.1476 |
| 1-month survival with CPC 1–2 | | 1,128 (21.1%) | 917 (21.4%) | 211 (20.0%) | 0.3331 |

Characteristics of the study population including fifteen predictor variables and an outcome were described. All categorical variables are shown as n (%). CAG, coronary angiography; CPC, cerebral performance category; CPR, cardiopulmonary resuscitation; EMS, emergency medical services; GCS M, motor response in the Glasgow coma scale; IABP, intra-aortic balloon pumping; PCI, percutaneous coronary intervention; TTM, target temperature management

Several reports have suggested that epinephrine increases the likelihood of ROSC; however, it does not affect survival in the long term and it may also worsen neurological prognosis [35, 36]. In this study, the absence of epinephrine was linked to survival with a good neurological prognosis. The first choice of treatment for shockable rhythm is defibrillation, and if ROSC is achieved immediately upon defibrillation, epinephrine is not administered. The prognosis is more favorable in patients with good responsiveness to defibrillation and short time to ROSC.

There are several limitations to this study. First, the Bayesian network analysis is a method unsuitable for continuous variables and the fact that only nominal variables were associated with a good prognosis in this study may be owing to the choice of analysis. Although the variables in this study were chosen from the existing literature, bias may be present in the selection of variables. Important underlying variables in addition to the variables used in the analysis are possible. Furthermore, the researchers specified the order of the nodes, which may have restricted the causal relationship [37]. Second, a total of 8,464 cases were excluded owing to missing data, and thus, the results may not be conclusive for the general population. Possible treatments and treatment protocols may differ depending on the participating facilities. The timing of the GCS M observation and blood sampling may differ among the patients, and the individual variables were not observed at a consistent time. In addition, the blood test measurements used in this study were a mixture of those taken immediately after ROSC and those taken on admission to the ICU after undergoing various treatments. Blood test results can change significantly before and after the treatment of cardiac arrest. Therefore, it may be desirable to standardize the timing of blood tests in all patients. Making comparisons with previously reported prognostic models was also difficult owing to the differences in collectible variables. Last, the interval between CRP initiation and ROSC was relatively short in some patients, making it difficult to evaluate whether these patients had a cardiac arrest. The possibility that some patients were erroneously diagnosed as patients with OHCA cannot be completely ruled out.

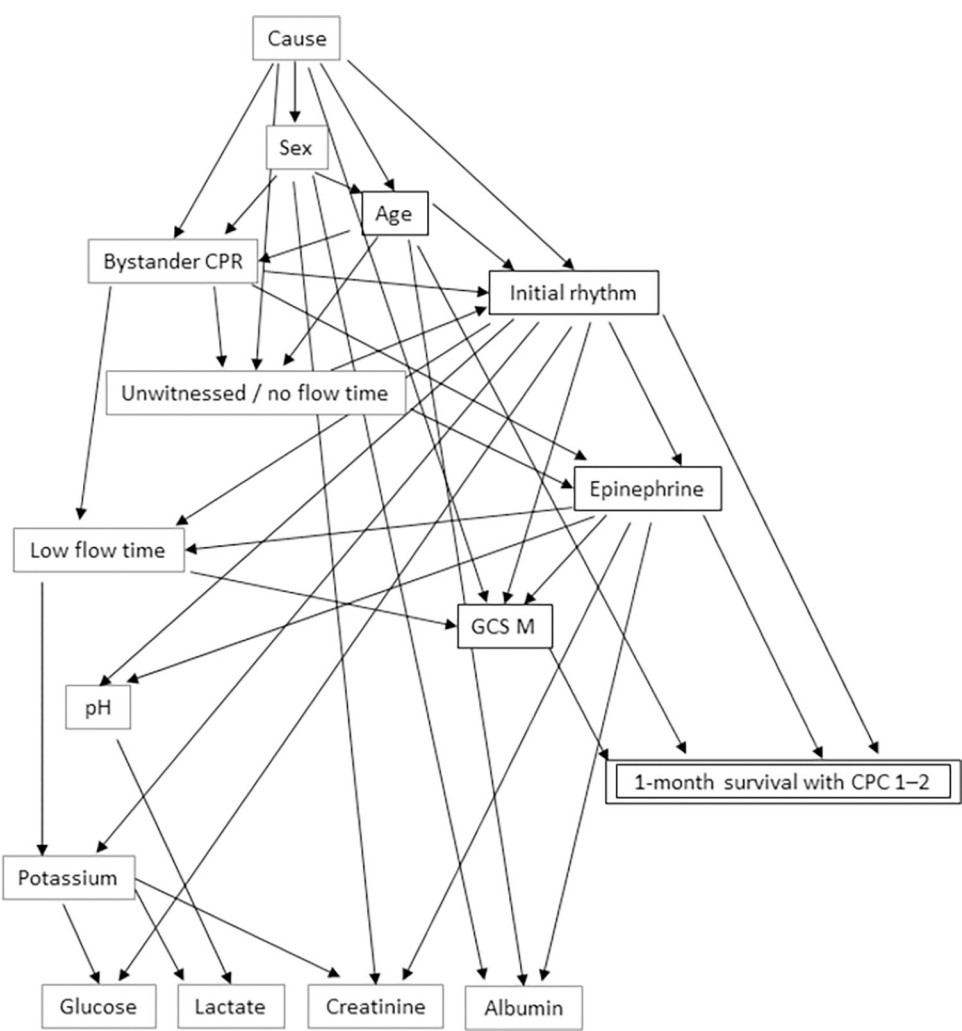

**Fig 2. Bayesian network by training set (2014–2018).** CPC, cerebral performance category; GCS M, motor response in the Glasgow coma scale.

**Table 2. Probability analysis.**

| Rank | Age | Initial rhythm | Epinephrine | GCS M | Probability value | Gap of probability values |
|------|-----|----------------|-------------|-------|-------------------|---------------------------|
| 1 | | | | 2–6 | 0.72 | 0.53 |
| 2 | | | Non-administered | | 0.62 | 0.43 |
| 3 | | Spontaneous rhythm | | | 0.48 | 0.29 |
| 4 | | Shockable | | | 0.45 | 0.26 |
| 5 | <58 | | | | 0.37 | 0.19 |
| 6 | 59–70 | | | | 0.25 | 0.06 |
| 7 | | | | | 0.19 | 0.00 |
| 8 | 71–80 | | | | 0.15 | -0.03 |
| 9 | | | | 1 | 0.13 | -0.06 |
| 10 | | Pulseless electrical activity | | | 0.10 | -0.09 |
| 11 | | | Administered | | 0.07 | -0.11 |
| 12 | >81 | | | | 0.07 | -0.12 |
| 13 | | Asystole | | | 0.03 | -0.16 |

GCS M, motor response in the Glasgow coma scale

**Table 3. The Bayesian model validation result.**

|  | Predict good outcome | Predict poor outcome | Sensitivity | Specificity | Positive predictive value | Negative predictive value |
|---|---|---|---|---|---|---|
| **Good outcome** | 159 | 39 | 75.4% | 95.4% | 80.3% | 94.0% |
| **Poor outcome** | 52 | 804 |  |  |  |  |

Good outcome: one-month survival with CPC 1–2; Poor outcome: none of one-month survival with CPC 1–2

## Conclusions

Using a Bayesian network, four variables, GCS M score of 2–6 after ROSC, initial rhythm (spontaneous rhythm and shockable), younger age, and absence of epinephrine were shown to be potentially closely associated with good neurological survival. These variables may help clinicians in their overall decision-making.

## Supporting information

**S1 Table. Estimated probability for one month survival with CPC 1–2.**
(XLSX)

**S2 Table. Variables in previous studies.**
(XLSX)

## Acknowledgments

The authors would like to acknowledge Editage (www.editage.com) for English language editing.

## Author Contributions

**Conceptualization:** Kota Shinada, Ayaka Matsuoka, Hiroyuki Koami, Yuichiro Sakamoto.

**Data curation:** Kota Shinada.

**Formal analysis:** Kota Shinada.

**Methodology:** Kota Shinada.

**Project administration:** Kota Shinada.

**Supervision:** Yuichiro Sakamoto.

**Validation:** Kota Shinada.

**Visualization:** Kota Shinada.

**Writing – original draft:** Kota Shinada, Ayaka Matsuoka.

**Writing – review & editing:** Kota Shinada, Ayaka Matsuoka, Hiroyuki Koami, Yuichiro Sakamoto.

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
