## [Decision Letter · Decision Letter 0]

2 May 2023

PONE-D-23-03455Bayesian network predicted variables for good neurological outcomes in patients with out-of-hospital cardiac arrestPLOS ONE

Dear Dr. Shinada,

Thank you for submitting your manuscript to PLOS ONE. After careful consideration, we feel that it has merit but does not fully meet PLOS ONE’s publication criteria as it currently stands. Therefore, we invite you to submit a revised version of the manuscript that addresses the points raised during the review process.

We look forward to receiving your revised manuscript.

Kind regards,

Gaetano Santulli

Academic Editor

PLOS ONE

Journal Requirements:

Reviewers' comments:

Reviewer's Responses to Questions

**Comments to the Author**

1. Is the manuscript technically sound, and do the data support the conclusions?

Reviewer #1: Partly

Reviewer #2: Partly

Reviewer #3: Partly

2. Has the statistical analysis been performed appropriately and rigorously? 

Reviewer #1: Yes

Reviewer #2: I Don't Know

Reviewer #3: Yes

3. Have the authors made all data underlying the findings in their manuscript fully available?

Reviewer #1: Yes

Reviewer #2: No

Reviewer #3: No

4. Is the manuscript presented in an intelligible fashion and written in standard English?

Reviewer #1: Yes

Reviewer #2: No

Reviewer #3: Yes

5. Review Comments to the Author

Reviewer #1: Major Point

The authors discussed about OHCA registry with Bayesian network.

To the readers of this paper, I guess that the explanation of machine learning and difference between Bayesian network and deep learning are insufficient.

Will this paper have more meanings more than that authors have used Bayesian network.

Please explain this point in detail more.

Reviewer #2: In this analysis, Shinada et al. aimed at finding and validating a subset of criteria for the prediction of survival at 1-month with good neurological outcome in OHCA patients. From a pool of fifteen variables selected by the authors based on previous publications, predictors of survival with a good neurological outcome were identified using a Bayesian network.

The authors found that 4 variables, namely age, initial rhythm, presence of epinephrine, and GCS M were associated with one-month survival with CPC 1–2.

The topic is interesting and the database large and apparently of good quality.

However, there are several major issues to be resolved:

1) the Registry and the methods of data collection should be better described: how many centers participate to the Japan Association for Acute 56 Medicine (JAAM) OHCA registry? What is the population of the area served by the facilities participating to the registry? Are the participating centers able to provide all the necessary treatments to OHCA patients? Please provide a more detailed description of the system as recommended by the Utstein criteria (https://doi.org/10.1161/CIR.0000000000000144) should be provided.

2) how was the clinical endpoint assessed? Was a study-specific follow-up conducted? or was survival and GCS assessed solely based on health records? How many patients had complete follow-up information?

3) The description of the Bayesian method should be much more detailed. Was a biostatistician involved?

4) It is a major limitation that continuous variables (eg. pH, glucose, creatinine) were categorized into quartiles resulting in cut-offs that have no relationship with validated cut-offs used in clinical activity. Probably a second method should be used to identify predictors of good neurological outcome.

5) A definition of intrinsic causes of cardiac arrest should be provided. Do the authors mean medical non traumatic causes?

6) The authors should provide the references, on which the selection of the 15 variables was based. It is a major limitation that the variables were selected based on previous literature and authors' judgement.

6) The results should be contextualized within the existing literature. In particular the authors should compare the findings of the current analysis with other available risk assessment tools (variables selected, sensitivity and specificity).

7) Additional information on baseline characteristics and interventions (e.g. CPR provided by emergency medical service, mechanical CPR, coronary angiography, PCI, therapeutic hypothermia), timing of death (KM curves) according to the Utstein criteria (https://doi.org/10.1161/CIR.0000000000000144) should be provided

8) English language should be improved. Examples:

- Line 99: “with a 100 dominance level set at 5%”: what do you mean?

- Line 107: “the patient background information”: i suggest "baseline characteristics and cardiac arrest details"

- Line 149: exacerbate the neurological prognosis: "worsen" or "affect" would be more appropriate

9) Please provide the measure units of the laboratory values.

10) Conclusions should be more informative: rather than "age, presence of prehospital adrenaline, GCS M after ROSC, and initial rhythm" i suggest to report older age, absence of prehospital adrenaline, type of rhythm (shockable vs. non shockable).

Reviewer #3: Comments

This study examined factors associated with good neurological outcome in patients with out-of-hospital cardiac arrest using a Bayesian network. Prediction of prognosis after resuscitation of patients with out-of-hospital cardiac arrest is a critical issue for treatment selection and explanation to patients' families. The strengths of this study are using large-scale data from the Japan Association for Acute Medicine OHCA registry and the All-Japan Utstein Registry of the Fire and Disaster Management Agency and using a Bayesian network.

On the other hand, the novelty of this study is somewhat weak, as several similar studies have been reported. In addition, although the authors identified variables that are particularly important in predicting the prognosis of out-of-hospital cardiac arrest, the significance of this identification is difficult to convey to the reader.

Major comments

41-43 Please describe the problems that should be improved in the prognostic and predictive scores of OHCA reported so far. Then, please describe your motivation for conducting this study to address these problems.

116 What is the basis or criteria for setting the values of No flow time and GCS M cutoffs in Table.1, please describe details in the statistic.

151-155 The authors discussed that some cases with ROSC after a relatively short CPR time were included; the prognosis may be better if ROSC does not require adrenaline, and it may also include patients in whom CPR was initiated because they were wrongly assessed to have an OHCA. However, no data are presented to show the relationship between low flow time and adrenaline administration. Also, data are not provided on how many patients were incorrectly evaluated for OHCA.

157-164 The authors discussed that the four variables identified using the Bayesian network may help clinicians in making decisions on the prognosis of OHCA and the overall treatment strategy. Please describe how you would like clinicians to use these four variables to make decisions.

Minor comments

67-69 It is better to have a light explanation of the meaning of no flow and low flow at the beginning because it is difficult for some readers to understand.

76 Please list citations for the 15 variables.

78 Please standardize the terminology to either adrenaline or epinephrine.

6. PLOS authors have the option to publish the peer review history of their article (what does this mean?). If published, this will include your full peer review and any attached files.

Reviewer #1: No

Reviewer #2: No

Reviewer #3: No

---

## [Author Response · Author response to Decision Letter 0]

12 Jul 2023

Reply to Reviewer 1’s comments

>We appreciate the reviewer’s comments. We have tried to incorporate the reviewer’s suggestions as much as possible, but welcome any additional comments that the reviewer may have.

The authors discussed about OHCA registry with Bayesian network.

To the readers of this paper, I guess that the explanation of machine learning and difference between Bayesian network and deep learning are insufficient.

Will this paper have more meanings more than that authors have used Bayesian network.

Please explain this point in detail more.

>We thank the reviewer for the suggestion. We have added an explanation in the Introduction regarding the significance of using Bayesian networks in this analysis, as well as a comparison with other deep learning methods (lines 39–53).

Replies to Reviewer 2’s comments

>We appreciate the reviewer’s comments. We have tried to incorporate the suggestions as much as possible, but welcome any additional comments that the reviewer may have.

1) the Registry and the methods of data collection should be better described: how many centers participate to the Japan Association for Acute 56 Medicine (JAAM) OHCA registry? What is the population of the area served by the facilities participating to the registry? Are the participating centers able to provide all the necessary treatments to OHCA patients? Please provide a more detailed description of the system as recommended by the Utstein criteria (https://doi.org/10.1161/CIR.0000000000000144) should be provided.

>We thank the reviewer for the suggestion. We have added a statement regarding the number of participating facilities (lines 61–62). However, it was difficult to confirm the population sample for the entire region. The registry includes a variety of hospitals and there may be differences in the care that can be provided. We have also added a statement stating the same in the limitations paragraph (lines 177).

2) how was the clinical endpoint assessed? Was a study-specific follow-up conducted? or was survival and GCS assessed solely based on health records? How many patients had complete follow-up information?

>We thank the reviewer for the question. There is no follow-up specific to the outcomes of this study. In Japan, one-month survival data is routinely collected in both the Fire and Disaster Management Agency Utstein Registry and Japan Association for Acute Medicine OHCA Registry. GCS and one-month survival was recorded by the respective centers; there were 5 missing cases regarding GCS M (Figure 1) and no cases with missing information on one-month survival.

3) The description of the Bayesian method should be much more detailed. Was a biostatistician involved?

>We have added a part on the Bayesian network method (lines 46–53). No biostatistician was involved in the study.

4) It is a major limitation that continuous variables (eg. pH, glucose, creatinine) were categorized into quartiles resulting in cut-offs that have no relationship with validated cut-offs used in clinical activity. Probably a second method should be used to identify predictors of good neurological outcome.

>We thank the reviewer for the suggestion. We have reanalyzed the blood test results by dividing them into categorical variables with clinically relevant values (lines 98–102).

5) A definition of intrinsic causes of cardiac arrest should be provided. Do the authors mean medical non traumatic causes?

>We thank the reviewer for pointing this out. We have replaced "intrinsic except cardiogenic" with "non-cardiogenic" because extrinsic cardiac arrest was excluded in the first place (Table 1).

6) The authors should provide the references, on which the selection of the 15 variables was based. It is a major limitation that the variables were selected based on previous literature and authors' judgement.

>We thank the reviewer for pointing this out. We have added some references (line 78) as per the reviewer’s suggestion. We have also mentioned the possibility of bias due to variable selection in the limitations (line 172–174).

6) The results should be contextualized within the existing literature. In particular the authors should compare the findings of the current analysis with other available risk assessment tools (variables selected, sensitivity and specificity).

>We thank the reviewer for the comment. We have added text on the comparison with previous studies for the selected variables in Table S2 . NULL-PLEASE scores are reported as excellent in previous systematic reviews (Gue YX, et al. Out-of-hospital cardiac arrest: A systematic review of current risk scores to predict survival. Am Heart J. 2021;234:31-41.). We attempted to compare the findings with the NULL-PLEASE score; however, the NULL-PLEASE score included variables that were not collected in the OHCA registry, making actual comparison difficult. We have added the lack of comparison with existing scores to the limitations paragraph (lines 182–184).

7) Additional information on baseline characteristics and interventions (e.g. CPR provided by emergency medical service, mechanical CPR, coronary angiography, PCI, therapeutic hypothermia), timing of death (KM curves) according to the Utstein criteria (https://doi.org/10.1161/CIR.0000000000000144) should be provided

>We thank the reviewer for pointing this out. We have added as much additional information as we could find: bystander defibrillation, physician-staffed EMS, ECPR, IABP, CAG, PCI, and TTM (Table 1). It was difficult to draw a KM curve for timing of death because of missing data.

8) English language should be improved. Examples:

- Line 99: “with a 100 dominance level set at 5%”: what do you mean?

- Line 107: “the patient background information”: i suggest "baseline characteristics and cardiac arrest details"

- Line 149: exacerbate the neurological prognosis: "worsen" or "affect" would be more appropriate

>We thank the reviewer for pointing this out.

- We have corrected the first statement to "P<0.05 was considered significant." (line 104-105)

- Corrected as suggested (lines 110–111).

- Corrected as suggested (line 154).

9) Please provide the measure units of the laboratory values.

>We thank the reviewer for the comment. We have added the unit of measure (Table 1).

10) Conclusions should be more informative: rather than "age, presence of prehospital adrenaline, GCS M after ROSC, and initial rhythm" i suggest to report older age, absence of prehospital adrenaline, type of rhythm (shockable vs. non shockable).

>We thank the reviewer for the comment. We have revised the conclusions as suggested (lines 25–26 and 189–190)

Reply to Reviewer 3’s comments

>We appreciate the reviewer’s comments. We have tried to incorporate the suggestions as much as possible, but welcome any additional comments the reviewer may have.

41-43 Please describe the problems that should be improved in the prognostic and predictive scores of OHCA reported so far. Then, please describe your motivation for conducting this study to address these problems.

>We thank the reviewer for the suggestion. We have added a description in the Introduction (lines 39–53).

116 What is the basis or criteria for setting the values of No flow time and GCS M cutoffs in Table.1, please describe details in the statistic.

>We thank the reviewer for pointing this out. Blood test items with reference values were divided by the reference value, but variables without reference values, such as the NFT and GCS M, were divided by quartiles. We have added corresponding text in the revised manuscript (lines 98–103).

151-155 The authors discussed that some cases with ROSC after a relatively short CPR time were included; the prognosis may be better if ROSC does not require adrenaline, and it may also include patients in whom CPR was initiated because they were wrongly assessed to have an OHCA. However, no data are presented to show the relationship between low flow time and adrenaline administration. Also, data are not provided on how many patients were incorrectly evaluated for OHCA.

>We thank the reviewer for the comment. As pointed out by the reviewer, there are no data to support this relationship; therefore, we have deleted the relevant text from the manuscript.

157-164 The authors discussed that the four variables identified using the Bayesian network may help clinicians in making decisions on the prognosis of OHCA and the overall treatment strategy. Please describe how you would like clinicians to use these four variables to make decisions.

>We thank the reviewer for the advice. We have added a part in the manuscript as per the suggestions (lines 163–169).

Minor comments

67-69 It is better to have a light explanation of the meaning of no flow and low flow at the beginning because it is difficult for some readers to understand.

>We thank the reviewer for the comment. We have added a brief explanation in the revised manuscript (lines 90–92).

76 Please list citations for the 15 variables.

>We thank the reviewer for pointing this out. We have added citations as suggested (line 78).

78 Please standardize the terminology to either adrenaline or epinephrine.

>We thank the reviewer for the comment. We have replaced “adrenaline” with "epinephrine" in each instance.

---

## [Decision Letter · Decision Letter 1]

28 Jul 2023

PONE-D-23-03455R1Bayesian network predicted variables for good neurological outcomes in patients with out-of-hospital cardiac arrestPLOS ONE

Dear Dr. Shinada,

Thank you for submitting your manuscript to PLOS ONE. After careful consideration, we feel that it has merit but does not fully meet PLOS ONE’s publication criteria as it currently stands. Therefore, we invite you to submit a revised version of the manuscript that addresses the points raised during the review process.

We look forward to receiving your revised manuscript.

Kind regards,

Gaetano Santulli, MD

Academic Editor

PLOS ONE

Journal Requirements:

Reviewers' comments:

Reviewer's Responses to Questions

**Comments to the Author**

1. If the authors have adequately addressed your comments raised in a previous round of review and you feel that this manuscript is now acceptable for publication, you may indicate that here to bypass the “Comments to the Author” section, enter your conflict of interest statement in the “Confidential to Editor” section, and submit your "Accept" recommendation.

Reviewer #1: All comments have been addressed

Reviewer #2: All comments have been addressed

Reviewer #3: All comments have been addressed

2. Is the manuscript technically sound, and do the data support the conclusions?

Reviewer #1: Yes

Reviewer #2: Yes

Reviewer #3: Yes

3. Has the statistical analysis been performed appropriately and rigorously? 

Reviewer #1: Yes

Reviewer #2: I Don't Know

Reviewer #3: Yes

4. Have the authors made all data underlying the findings in their manuscript fully available?

Reviewer #1: Yes

Reviewer #2: Yes

Reviewer #3: Yes

5. Is the manuscript presented in an intelligible fashion and written in standard English?

Reviewer #1: Yes

Reviewer #2: Yes

Reviewer #3: Yes

6. Review Comments to the Author

Reviewer #1: Previous comment is replied appropriately about the explanation of machine learning and difference between Bayesian　network and deep learning. I think that this paper is acceptable.

Reviewer #2: I thank the authors for addressing the majority of my concerns.

I have few additional comments:

1) Line 24: I suggest to replace “non-cardiac arrest” with “spontaneous rhythm” or “ROSC”; the label “non-cardiac arrest” as type of rhythm is very confusing.

2) Table 1: I suggest to replace “cardiogenic and non cardiogenic cause” with “cardiac or noncardiac” in agreement with the Utstein guidelines (https://doi.org/10.1161/CIR.0000000000000144)

3) The authors answered to a previous comment: “There is no follow-up specific to the outcomes of this study. In Japan, one-month survival data is routinely collected in both the Fire and Disaster Management Agency Utstein Registry and Japan Association for Acute Medicine OHCA Registry. GCS and one-month survival was recorded by the respective centers; there were 5 missing cases regarding GCS M (Figure 1) and no cases with missing information on one-month survival.”

This information should be reported in the methods.

4) Please mention in the discussion previous studies that assessed the sensitivity and specificity of other scores, such as the NULL PLEASE

Reviewer #3: The author generally answered all questions somewhat adequately and the arguments were easy to understand.

7. PLOS authors have the option to publish the peer review history of their article (what does this mean?). If published, this will include your full peer review and any attached files.

Reviewer #1: No

Reviewer #2: No

Reviewer #3: No

---

## [Author Response · Author response to Decision Letter 1]

23 Aug 2023

August 24 2023

Emily Chenette

Editor-in-Chief

PLoS One

Dear Editor-in-Chief:

We thank you for considering our paper titled “Bayesian network predicted variables for good neurological outcomes in patients with out-of-hospital cardiac arrest,” manuscript ID, PONE-D-23-03455R1, for publication in PLoS One. We are hereby re-submitting the revised version of our manuscript.

We are grateful for the feedback provided by the reviewers, which has helped improve the quality of our paper. In accordance with the valuable comments, we have made corrections and additions to the text and tables.

The corrections and additions in the text are highlighted in yellow in the revised manuscript.

We have tried to incorporate your suggestions as much as possible; however, if you have any further suggestions, please let us know.

Thank you for your constructive comments.

Sincerely,

Kota Shinada

Department of Emergency and Critical Care Medicine, 

Faculty of Medicine, Saga University

5-1-1 Nabeshima

Saga City, Saga Prefecture 849-8501, Japan

Phone number: +81-952-34-3160

Fax number: +81-952-34-1061

Email address: st9137@cc.saga-u.ac.jp  

Reply to Reviewer 1’s comments

Previous comment is replied appropriately about the explanation of machine learning and difference between Bayesian　network and deep learning. I think that this paper is acceptable.

Response: Thank you for your confirmation and comments.

Reply to Reviewer 2’s comments

I thank the authors for addressing the majority of my concerns.

I have few additional comments:

Response: Thank you very much for your confirmation and additional constructive comments.

1) Line 24: I suggest to replace “non-cardiac arrest” with “spontaneous rhythm” or “ROSC”; the label “non-cardiac arrest” as type of rhythm is very confusing.

Response: We thank the reviewer for the suggestion. We have accordingly made corrections to the lines. (Lines 24, 26, 139, and 196 and tables 1, 2, and S1)

2) Table 1: I suggest to replace “cardiogenic and non cardiogenic cause” with “cardiac or noncardiac” in agreement with the Utstein guidelines (https://doi.org/10.1161/CIR.0000000000000144)

Response: We thank the reviewer for the suggestion. We have accordingly made the corrections. (Table 1)

3) The authors answered to a previous comment: “There is no follow-up specific to the outcomes of this study. In Japan, one-month survival data is routinely collected in both the Fire and Disaster Management Agency Utstein Registry and Japan Association for Acute Medicine OHCA Registry. GCS and one-month survival was recorded by the respective centers; there were 5 missing cases regarding GCS M (Figure 1) and no cases with missing information on one-month survival.”

This information should be reported in the methods.

Response: We thank the reviewer for pointing this out. We have included this in the Methods section. (Lines 72-74)

4) Please mention in the discussion previous studies that assessed the sensitivity and specificity of other scores, such as the NULL PLEASE

Response: We thank the reviewer for the suggestion. Citing a systematic review, we have mentioned the predictive performance of the NULL-PLEASE score in the Discussion section. However, it was difficult to mention the sensitivity and specificity as they were not mentioned in most previous studies, although AUC was mentioned.

Reply to Reviewer 3’s comments

The author generally answered all questions somewhat adequately and the arguments were easy to understand.

Response: Thank you for your confirmation and comments.

---

## [Editor Report · Decision Letter 2]

25 Aug 2023

Bayesian network predicted variables for good neurological outcomes in patients with out-of-hospital cardiac arrest

PONE-D-23-03455R2

Dear Dr. Shinada,

We’re pleased to inform you that your manuscript has been judged scientifically suitable for publication and will be formally accepted for publication once it meets all outstanding technical requirements.

Kind regards,

Gaetano Santulli, MD

Academic Editor

PLOS ONE

---

## [Editor Report · Acceptance letter]

18 Sep 2023

PONE-D-23-03455R2 

Bayesian network predicted variables for good neurological outcomes in patients with out-of-hospital cardiac arrest 

Dear Dr. Shinada:

I'm pleased to inform you that your manuscript has been deemed suitable for publication in PLOS ONE. Congratulations! Your manuscript is now with our production department. 

Kind regards, 

on behalf of

Professor Gaetano Santulli 

Academic Editor

PLOS ONE